# Impact of the COVID-19 Pandemic on Hemato-Oncology Services: A Retrospective Dual-Center Cohort Study in Kazakhstan

**DOI:** 10.3390/healthcare13192520

**Published:** 2025-10-04

**Authors:** Maral Yerdenova, Aigulsum Izekenova, Akbope Myrkassymova, Gaukhar Mergenova, Mohammed Merzah, Balday Issenova, Maksat Mamyrkul, Aliya Atabayeva, Vytenis Kalibatas, Dejan Nikolic, Yineng Chen

**Affiliations:** 1Department of Epidemiology with the Course of HIV Infection and IK, Asfendiyarov Kazakh National Medical University, Almaty 050012, Kazakhstan; erdenova.maral@kaznmu.kz (M.Y.); mergenova.g@kaznmu.kz (G.M.); 2Department of Biostatistics and Basics of Scientific Research, Asfendiyarov Kazakh National Medical University, Almaty 050012, Kazakhstan; myrkassymova.a@kaznmu.kz; 3Global Health Research Center of Central Asia, Columbia University School of Social Work, Almaty 050012, Kazakhstan; 4Department of Public Health and Epidemiology, Faculty of General Medicine, University of Debrecen, 4028 Debrecen, Hungary; merzah.karbala@gmail.com; 5Department of Community Health, Technical Institute of Karbala, Al-Furat Al-Awsat Technical University, Kufa 54003, Iraq; 6Department of Nursing, Asfendiyarov Kazakh National Medical University, Almaty 050012, Kazakhstan; isenova.b@kaznmu.kz; 7Department of Health Politics and Management, Asfendiyarov Kazakh National Medical University, Almaty 050012, Kazakhstan; maksat333777@gmail.com; 8Department of Public Health, NJSC Semey Medical University, Semey 071400, Kazakhstan; aliya.atabayeva@smu.edu.kz; 9Department of Health Management, Lithuanian University of Health Sciences, 44307 Kaunas, Lithuania; vytenis.kalibatas@gmail.com; 10Faculty of Medicine, University of Belgrade, 11000 Belgrade, Serbia; denikol27@gmail.com; 11Department of Physical Medicine and Rehabilitation, University Children’s Hospital, 11000 Belgrade, Serbia; 12College of Integrated Health Science, University at Albany, Albany, NY 12144, USA; ychen77@albany.edu

**Keywords:** healthcare services, COVID-19, pandemics, hemato-oncology care, access to healthcare, healthcare delivery

## Abstract

**Background:** Numerous healthcare services have been affected by the COVID-19 pandemic worldwide. Specialized healthcare services were postponed or canceled, potentially compromising regular services for hemato-oncology patients. The current study aimed to analyze the impact of the COVID-19 pandemic on access to hemato-oncology services in Almaty, the largest city in Kazakhstan. **Methods:** We retrospectively analyzed the socio-demographic characteristics of patients admitted to two large tertiary centers rendering hemato-oncology services, the City Clinical Hospital 7 (H7) and the Kazakh Institute of Oncology and Radiology (KazIOR). All data were retrieved for the period spanning from 1 March 2019 to 28 February 2022. The retrieved variables included age, gender, type of residence, hospitalization rate, treatment outcomes (discharged/deceased), bed days, diagnoses according to International Classification of Diseases (ICD-10) (acute leukemia and hematopoietic depression, lymphoproliferative diseases, and myeloproliferative diseases), and referral sources (ambulance, another hospital, consultative and diagnostic assistance, primary healthcare, self-referral, and referrals from hematologists’ offices). **Results:** In the 2019–2022 period, 6763 hemato-oncology hospitalizations were registered: 3583 in H7 and 3180 in KazIOR. The mean age at hospitalization was 55.04 (SD = 16.07) for females and 51.2 (SD = 16.7) for males. The proportion of hospitalized urban and rural patients differed significantly: 6191 (92%) and 571 (8,4%), respectively (χ^2^ = 13.8, *p* = 0.001). In the 2020–2021 period, fewer patients were discharged (n = 2047) compared to 2019–2020 (n = 2387) and 2021–2022 (n = 2081) (χ^2^ = 20.09, *p* = 0.003). However, the proportion of deaths in the 2020–2021 period (3.5%) was higher than in the 2019–2020 (3.2%) and 2021–2022 periods (2.6%) (χ^2^ = 20.09, *p* = 0.003). A total of 403 (19%) hospital admissions were carried out by ambulance (emergency cases) in the 2020–2021 period, 368 (14.8%) in 2019–2020, and 394 (18.3%) in 2021–2022 (χ^2^ = 2231, *p* < 0.001). The number of patients transferred from other hospitals to H7 and KazIOR increased by 12.4% in the 2020–2021 period. **Conclusions:** Our findings indicate a negative impact of the COVID-19 pandemic on access to hemato-oncology services, leading to increased mortality. Further studies are warranted to explore factors underlying the trends in hospitalizations and mortality of hemato-oncology patients during healthcare crises.

## 1. Introduction

The COVID-19 pandemic has strained healthcare systems worldwide, disrupting the coordination between primary, secondary, and tertiary services [1,2]. Hematology and oncology services for patients with immunocompromised conditions have been particularly affected [3]. The pandemic has necessitated adaptations in healthcare delivery for hemato-oncology patients to prevent the deaths of patients suffering from a lack of essential drug supply and postponed hospitalizations [4,5].

Hemato-oncology patients were especially vulnerable during the pandemic due to uncertainties of COVID-19’s impact on disease progression and treatment outcomes, the absence of recommendations tailored to the needs of patients depending on the type of hemato-oncologic diagnosis, and the severity of disease [3,6]. International registries such as EPICOVIDEHA and CCC19 have demonstrated significantly increased morbidity and mortality in patients with hematologic malignancies compared to the general cancer population [5,7]. Recent regional studies in Kazakhstan have also highlighted similar disruptions in oncology services during the pandemic [8].

Attempts to use innovative technologies, including telemedicine and telehealth consultations, were largely limited to developed urban areas and were unavailable in remote and rural areas. Consequently, healthcare providers were challenged with developing evidence-based strategies to ensure adequate care of hemato-oncology patients [9,10]. These global and regional disparities highlight the challenges in maintaining continuity of care, which are especially pronounced in countries with limited healthcare infrastructure [11].

In Kazakhstan, the first cases of COVID-19 were reported in March 2020 [12], marking the beginning of an unprecedented public health challenge. Similar to other countries, the pandemic amplified existing gaps in healthcare access, particularly for specialized services such as hemato-oncology care. Quarantine measures were quickly enforced to prevent the spread of the virus. Shortages of drug supplies, hospital beds, and qualified healthcare personnel, along with the provision of substandard care, raised significant concerns among local specialists [7,13,14].

Meanwhile, the global burden of hematologic malignancies remains substantial. The incidence of hemoblastosis, also known as blood cancer, is increasing annually worldwide [15]. For patients diagnosed with blood cancers, specialized hematology and oncology centers provide various services, including diagnostic evaluations such as complete blood count, bone marrow biopsy, and cytogenetic analysis. Treatment options typically involve chemotherapy, targeted therapy, immunotherapy, and, in some cases, hematopoietic stem cell transplantation. Supportive care services are crucial to managing complications like infections, anemia, and bleeding disorders. Regular follow-up and monitoring are essential for assessing treatment response and managing potential relapses. Globally, over 467,000 new cases of leukemia are reported each year, resulting in approximately 310,000 deaths [15]. In Kazakhstan, blood oncology diseases account for about 5% of all cancer cases, with approximately 1400 new cases annually and 7000 prevalent cases (based on 2022 recorded data) [16].

Although some studies have been conducted globally, research on how public health crises such as the COVID-19 pandemic affect access to hemato-oncology services in Central Asia remains limited. This study aims to describe access to and the state of healthcare services for hemato-oncology patients before and during the COVID-19 pandemic in two major healthcare centers in Almaty, the largest city in Kazakhstan, providing insights into the local impact of the pandemic on this vulnerable population.

## 2. Methods

### 2.1. Study Design and Data Collection

A retrospective dual-center study analyzed data from patients admitted to two major tertiary centers providing hemato-oncology services, City Clinical Hospital No. 7 (H7) and the Kazakh Institute of Oncology and Radiology (KazIOR), both of which exclusively deliver this care at the city level. H7 provides both elective and emergency hospitalizations, while KazIOR admits patients for planned cases only. H7 is a multidisciplinary hospital with a capacity of 662 beds, providing a wide range of medical services, including adult hematological care at the tertiary healthcare level. KazIOR is a national scientific research and referral institute at the tertiary healthcare level, specializing in oncology and radiology, with a total of 356 beds. Data were collected from 1 March 2019 to 28 February 2022.

### 2.2. Study Setting

Data were extracted from the Integrated Medical Information System (IMIS), an electronic database designed for archiving documentation, available for searches to authorized users in Kazakhstan [17]. The database was rigorously de-identified. All identifiable information was removed, and each unique record was assigned a specific code to ensure maximum privacy and data integrity. We extracted data depending on hospitalization date, from March 2019 to February 2020—the pre-pandemic period; from March 2020 to February 2021—Wave 1; and from March 2021 to February 2022—Wave 2. There were no missing data for the tested parameters, and to ensure no coding errors occurred during the data extraction, two researchers (M.Y. and A.I.) performed a de-identification process and specific code placement.

### 2.3. Measures

We extracted the following characteristics of the patients: socio-demographic variables, including age (divided into groups: below 18 years, 18–29, 30–44, 45–59, above 60); sex; type of residence (urban or rural); the outcome of treatment (discharged or deceased); type of hospitalization (emergency/planned); referral sources, including from another hospital, consultative and diagnostic care (CDC), primary healthcare (PHC), patient self-referral, admission by ambulance, and referrals from the hematologists’ office; and bed days.

We also collected data about the diagnoses of patients who visited these two hospitals. Diagnoses were classified according to the International Classification of Diseases (ICD-10) and combined according to the Clinical-Cost Group (CCG, a system developed by the Ministry of Health Care of Kazakhstan to group patients based on similar clinical characteristics and treatment costs) [18]. The primary registered diagnoses were as follows: acute leukemia, anemia, hemolytic anemia, lymphoproliferative disorders, malignant disorders, myeloproliferative disorders, other neoplasms, other peripheral diseases, and pathology of hemostasis. Patient hospitalizations with non-hemato-oncology diseases were excluded from the analyses.

All processed data were also available from the Statistical Yearbook of the Republic of Kazakhstan. The Local Ethics Committee of Asfendiyarov Kazakh National Medical University approved this study (protocol No. 1217 dated 28 October 2021).

### 2.4. Statistical Analysis

A descriptive analysis was performed. Continuous variables were reported as means ± standard deviations (SD), and categorical variables were expressed as percentages. Chi-square or Fisher’s exact test was used to compare categorical variables between groups. The normal distribution of continuous variables was analyzed using the Kolmogorov–Smirnov test. Depending on whether the distribution was normal or not, we performed either Student’s *t*-test or the Mann–Whitney U test. An ANOVA test was used to evaluate the statistical difference in evaluated parameters among more than two tested time periods. A two-tailed *p*-value of <0.05 was considered statistically significant. Data analysis was conducted using SPSS software (IBM SPSS Statistics for Windows, Version 25, Armonk, NY, USA).

## 3. Results

### 3.1. Socio-Demographic

Between 2019 and 2022, 6763 hemato-oncology hospitalizations were recorded across two hospitals, H7 and KazIOR, encompassing both the pre-pandemic period and the two waves of the COVID-19 pandemic. Frequencies of patients between the pre-pandemic period and Wave 1 and Wave 2 periods significantly differed between different age categories (*p* < 0.001), area of residence (*p* = 0.002), hospitalization type (*p* < 0.001), and hospitalization outcome (*p* = 0.003), while a non-significant distribution was noticed for gender distribution (*p* = 0.450) (Table 1). The female gender was more frequent in all tested time periods; nine out of ten of all patients belonged to the group from urban areas, around two-thirds of patients had a planned type of hospitalization, and more than nine out of ten patients were discharged as a hospitalization outcome. Patients aged over 60 years represented the most frequent group in all three tested periods. The proportion of emergency hospitalizations increased between the Wave 1 (31.8%) and Wave 2 (37.0%) pandemic periods, with an initial decline between the pre-pandemic (33.6%) and Wave 1 pandemic periods. In contrast, planned hospitalizations decreased between Wave 1 (68.2%) and Wave 2 (63.0%) periods, with an initial increase between pre-pandemic (66.4%) and Wave 1 pandemic periods.

Gender distribution significantly differed between the pre-pandemic period and different pandemic periods (Wave 1 and Wave 2) for patients with emergency hospitalization type (*p* = 0.002); the female distribution for emergency hospitalization increased from the pre-pandemic period to the period of Wave 2. Non-significant distribution between different pandemic periods was noticed between genders for planned hospitalization type (*p* = 0.481) (Table 2).

### 3.2. Referral Source

Considering referral sources, the frequencies of patients between the pre-pandemic and Wave 1 and Wave 2 pandemic periods significantly differed for another hospital referral source (*p* < 0.001), consultative and diagnostic care (*p* < 0.001), primary healthcare (*p* < 0.001), self-referral (*p* < 0.001), and referrals from hematologists’ offices (*p* < 0.001), while non-significant distribution was observed when the ambulance was the referral source (*p* = 0.672) (Table 3).

Self-referrals demonstrated the most substantial fluctuation, with a decrease of 42.94% during Wave 1 and a decline of 6.47% in Wave 2 compared to the pre-pandemic period. Similarly, referrals from hematologists’ offices exhibited sharp declines of 88.1% and 92.32% in Wave 1 and Wave 2, respectively.

Consultative and diagnostic services, as well as primary healthcare referrals, showed contrasting patterns. While consultative and diagnostic referrals increased substantially (401.26% in Wave 1 and 437.11% in Wave 2), primary healthcare referrals decreased (−25.98% in Wave 1 and −30.91% in Wave 2).

Referrals from another hospital markedly increased from the pre-pandemic period, with 15 cases to a 1726.67% change in Wave 1 and a 1566.67% change in Wave 2 pandemic periods, suggesting a shift in the healthcare network’s referral dynamics. Conversely, ambulance referrals showed only a slight increase of 5.61% for Wave 1 and 5.88% for Wave 2 pandemic periods compared to the pre-pandemic period.

### 3.3. Hospitalization Days and Number of Visits

Hospitalization patterns for hemato-oncology patients shifted notably across the pre-pandemic, Wave 1, and Wave 2 periods, with clear changes in both the duration and distribution of hospital stays. A sharp peak in hospitalization for 7 days was evident in all three periods, most prominently during Wave 2, which exceeded 300 cases. A secondary, smaller peak was observed near 14 days of hospitalization across all groups. Hospital stays during Wave 1 were predominantly shorter, with most discharges occurring within the first 10 days. In contrast, the pre-pandemic period showed a lower frequency of hospitalizations and a wider range of durations, while Wave 2 exhibited an intermediate pattern in both peak frequency and spread.

Our analysis revealed significant differences in the mean duration of hospitalization across the three study periods. Hospital stays were significantly shorter during Wave 1 compared to the pre-pandemic period (Z = −14.40, *p* < 0.001), and even further reduced during Wave 2 (Z = −17.60, *p* < 0.001). A significant difference was also observed between Wave 1 and Wave 2, with hospitalizations being shorter in Wave 2 (Z = −2.82, *p* = 0.005) (Figure 1). These findings indicate a progressive reduction in the length of hospital stays as the pandemic evolved.

The distribution of repeated visits per patient varied across the pre-pandemic and pandemic periods, as illustrated in Figure 2. Across all timeframes, the majority of patients had between 1–2 and 3–5 visits, with frequencies declining steadily as the number of visits increased. Notably, the pre-pandemic period consistently showed higher frequencies of repeated visits compared to both pandemic waves, particularly in the categories of 1–2 and 3–5 visits. Wave 1 exhibited a slight reduction in repeated visits, while Wave 2 demonstrated the lowest frequencies across nearly all visit ranges. The association analysis indicates a notable shift toward more frequent visits during the pre-pandemic period (χ^2^ = 486.23, df = 20, *p* < 0.001).

Table 4 presents the mean numbers of visits and bed days for various final diagnoses across the three periods (pre-pandemic, Wave 1, and Wave 2). Overall, there was a noticeable decline in both metrics during the pandemic waves compared to the pre-pandemic baseline.

Patients diagnosed with acute leukemia, anemia, hemolytic anemia, and lymphoproliferative disorders showed relatively stable numbers of visits across the periods, with only slight decreases in Wave 2. However, the average number of bed days declined more substantially, especially for acute leukemia (from 20 days pre-pandemic to 14 days in Wave 2) and hemolytic anemia (from 15 to 10 days).

In contrast, malignant disorders exhibited a sharp reduction in visits, from 16 visits pre-pandemic to 5 and 4 during Waves 1 and 2, respectively, while bed days remained consistently lower across all periods. Myeloproliferative disorders and other neoplasms also followed this trend, with fewer visits and shorter hospital stays during the pandemic, particularly in Wave 2.

For rarer conditions such as pathology of hemostasis, a significant drop in both visits and bed days was observed in Wave 2. Data for other peripheral arterial diseases were incomplete but showed a spike in visits during Wave 1, with a corresponding reduction in bed days.

Statistical analysis revealed a highly significant difference in the number of visits across the pre-pandemic and pandemic periods (*p* < 0.001), and a statistically significant reduction in bed days (*p* = 0.034) (Table 4).

### 3.4. Hospitalization Outcomes

Table 5 presents the patient outcomes, categorized as death or discharge, across three periods (pre-pandemic, Wave 1, and Wave 2) in H7 and KazIOR.

Mortality rates varied across diagnoses but were not statistically significant (*p* = 0.300). Patients with acute leukemia consistently had the highest mortality, peaking at 40.3% during Wave 1 before declining to 21.8% in Wave 2. Mortality for anemia and myeloproliferative disorders showed fluctuations, with anemia peaking at 50.0% during Wave 2 and myeloproliferative disorders rising again to 40.0% in the same period.

Discharge outcomes showed significant variation across periods (*p* < 0.001). Patients with acute leukemia experienced a decrease in discharge rates from 39.2% pre-pandemic to 29.7% during Wave 1, slightly increasing to 31.2% in Wave 2. Similar declines were observed in lymphoproliferative and myeloproliferative disorders (Table 5).

## 4. Discussion

The COVID-19 pandemic has posed unprecedented challenges to healthcare systems worldwide, significantly impacting the delivery of hemato-oncology services. Our retrospective dual-center cohort study provides valuable insights into how two major hospitals that provided specialized hemato-oncologic care for the country’s biggest city adapted their practices to manage hemato-oncology patients across three distinct periods: pre-pandemic, Wave 1, and Wave 2.

An age shift was observed, with a growing proportion of hospitalized patients aged over 60 years during the pandemic waves. This aligns with global evidence indicating that older adults were more vulnerable to both COVID-19 and disruptions in chronic disease management. This shift may reflect the vulnerability to COVID-19 of older individuals with hemato-oncology, which is in line with previous cohort studies [19,20]. Research conducted in Mexico found that patients aged ≥70 years with hematologic malignancies had a significantly higher risk of mortality from COVID-19 [9].

The overwhelming majority of hospitalizations were among urban residents, likely reflecting better access to specialized hemato-oncology services in urban centers. Although rural patient numbers increased slightly, the persistent urban–rural gap underscores potential barriers to care in rural areas.

Emergency hospitalizations rose significantly during Wave 2, suggesting that delays in accessing planned or preventive care during earlier waves may have led to more acute presentations later in the pandemic. The level of emergency hospitalizations of hemato-oncology patients may indicate disruption of the regular care of these patients, which may have caused significant worsening of the course of the disease and the need for emergency hospitalization. This trend underscores the critical need for timely interventions and the strain on hospital resources to accommodate surges in emergency patient numbers.

The proportion of mortality in our study was higher during the pandemic (3.5% in Wave 1). The increased mortality could be due to a strained healthcare system and the indirect effects of suffering from COVID-19. Importantly, a study by the European Hematology Association revealed a devastating effect on hemato-oncology patients’ exposure to COVID-19, with 31% (1185 patients with COVID-19) reported to be dead at the beginning of the pandemic [7]. Overall, the risks of severe COVID-19 and related deaths were higher in a wide variety of oncology patients during the pandemic [21]. This clearly indicates the need for the timely proposal and implementation of adequate preventive measures, particularly related to COVID-19 and its consequences during the COVID-19 pandemic. The positive impact of vaccination was reported by Khatrawi and Sayed, who stated that upon vaccination in Saudi Arabia, there was a significant reduction in the percentage of daily COVID-19 cases [22].

Referrals by ambulance to specialist hemato-oncology services were proportionally high during Waves 1 and 2. These findings may indicate that hemato-oncology patients experienced delayed or canceled visits to receive necessary treatment. However, postponed visits can lead to worsening health conditions, ultimately requiring emergency hospitalization, which necessitates calling an ambulance. Our findings are different from a Korean study examining the impact of the pandemic on hemato-oncology care in a tertiary hospital, which reported an 18% decrease in ED visits by cancer patients, suggesting that emergency care of cancer patients was jeopardized during the same period [1].

At the same time, we found that the number of patients referred to hemato-oncology healthcare facilities (H7 and KazIOR) from other hospitals significantly increased during Waves 1 and 2 compared to the pre-pandemic period, and the majority of hemato-oncology patients in H7 and KazIOR were referred by the hematologist at the primary health care facility, which indicates that regular access to diagnostic and treatment services at primary health care facilities was disrupted for hemato-oncology patients during the pandemic. This disruption resulted in the hospitalization of hemato-oncology patients in other hospitals in Almaty and other regions in Kazakhstan, which could not provide them with specialized care. In a systematic review by Riera et al., it was stated that there was a reduction in routine cancer services activities, including visits, as well as a reduction in cancer surgeries and radiotherapy delays, along with cancellations, rescheduling, and delays in outpatient visits of cancer patients during COVID-19 [23]. All of these factors could be potential explanations for an increase in emergency hospitalizations. Radhakrishnan et al., authors from India, reported a significant drop in outpatient services for adult and pediatric hematology patients, as well as a drop in daily care of one patient [24]. Furthermore, the authors reported a 50% drop in inpatient bed occupancy, with pediatric services initially being unchanged [24].

The increase in emergency admissions in our study emphasizes the need for hospitals to enhance their emergency response capabilities, including critical care capacity and adequate staffing. This can be achieved through strategic planning and by training healthcare professionals. In a study from Saudi Arabia, the negative consequences of the COVID-19 pandemic, particularly on the delay in the diagnosis and management of most cancer patients, were reported; however, guidelines and the implementation of preventive measures had a positive impact on the reduction in delays and improved cancer care delivery [25]. This clearly demonstrates the importance of early monitoring of changes during pandemic periods and the need to better understand their influences on numerous aspects of the healthcare system, including diagnosis, management, and follow-up of patients, bearing in mind sensitive individuals. Multidisciplinary and interdisciplinary coordinated cooperation will lead to prompt and timely responses in terms of strategies and measures that should be proposed and implemented to improve disrupted delivery of healthcare services and ultimately optimize patient care.

The reliance on diverse referral systems at our hospitals highlights the importance of coordinated care during crises. Additionally, strategic planning of specialist training and resource allocation is needed to meet the demands of the current and future pandemics [26].

The shift towards older patient populations necessitates targeted strategies to protect and manage these individuals. Hemato-oncology hospitals should implement robust infection control measures, prioritize vaccinations for older patients, and develop specialized care protocols to address these complex needs.

To enhance the resilience of hemato-oncology services in the face of future pandemics, complex strategies should be developed. Prioritizing telehealth procedures can enable improved coordinated care of hemato-oncology patients, minimizing in-person visits to hospitals and reducing infection risks. Hemato-oncology hospitals may develop plans to rationalize emergency services, mobilize multidisciplinary teams, and expand online services during crises. Clinical protocols should be revised because of the pandemic experience worldwide [27,28].

Furthermore, ongoing research is needed to evaluate the long-term impact of the COVID-19 pandemic on hemato-oncology outcomes. Studies should focus on the effectiveness of adaptive strategies implemented during the pandemic and identify best practices that can be integrated into standard care protocols.

## 5. Limitations

This study has several limitations. First, we only relied on data from two major hospitals, where most hemato-oncology patients are registered, and data from other tertiary hospitals providing hemato-oncology services in Kazakhstan were not included. Second, the findings reflect the experience of a single country (Kazakhstan); therefore, multicenter studies across different countries are needed to capture a broader range of healthcare contexts, measures, and patient characteristics. Third, potential sources of bias should be acknowledged, including unmeasured factors such as educational level, income, living conditions, barriers to healthcare referral, and type of services accessed. Therefore, the generalization of our results is limited.

## 6. Conclusions

Our retrospective dual-center cohort study provides valuable insights into the changes and challenges faced by hemato-oncology services during the COVID-19 pandemic. We observed an increase in emergency hospitalizations from Wave 1 to Wave 2 of the pandemic, following an initial slight decline in Wave 1 compared with the pre-pandemic period. In contrast, planned hospitalizations and mortality frequency declined from Wave 1 to Wave 2, after showing a modest increase in Wave 1 compared with the pre-pandemic period. Referral patterns also shifted, with the greatest increase coming from other hospitals, while referrals from hematologists’ offices showed the most notable decline. Hospitalization peaks were most frequently observed on the seventh day, and the majority of patients were admitted one to two times.

The findings highlight the critical need for emergency preparedness and targeted care for older and vulnerable populations. By learning from these experiences and implementing strategic improvements, healthcare systems can better safeguard hemato-oncology patients and ensure the continuity of essential services during future crises.

## Figures and Tables

**Figure 1 healthcare-13-02520-f001:**
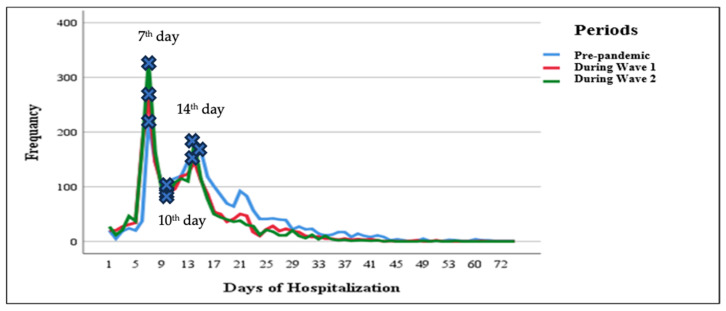
Hospitalization days across the pre-pandemic and pandemic periods. The figure presents the frequencies of pre-pandemic and pandemic periods in different days of hospitalization. The highest first two peaks and the lowest first peak were described with marked peaks for the pre-pandemic and pandemic periods at three different time points.

**Figure 2 healthcare-13-02520-f002:**
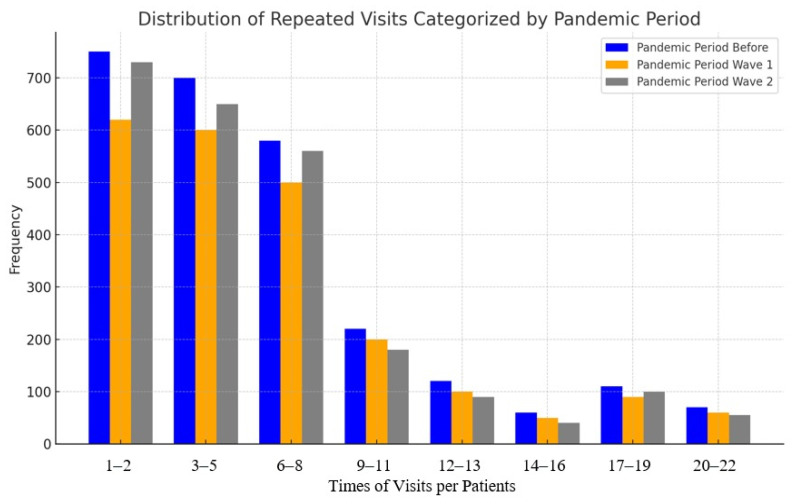
Number of visits categorized by the pre-pandemic and pandemic periods. The figure presents the frequencies of visits in the pre-pandemic and pandemic periods per patient. The bars of the pre-pandemic and pandemic periods are in groups with defined times of visits per patient.

**Table 1 healthcare-13-02520-t001:** Socio-demographic characteristics of the study population for all visits (N = 6763).

Variable		Pandemic Period			*p* *
		Pre-Pandemic (n, %)	Wave 1 (n, %)	Wave 2 (n, %)	
Age Categories					
	<17	13 (0.5)	4 (0.2)	0 (0.0)	<0.001
	18–29	307 (12.0)	190 (9.1)	189 (8.9)	
	30–44	494 (19.3)	415 (19.8)	389 (18.4)	
	45–59	732 (28.7)	613 (29.2)	593 (28.1)	
	>60	1008 (39.5)	875 (41.7)	941 (44.6)	
Gender					
	Female	1420 (55.6)	1185 (56.5)	1213 (57.4)	0.450
	Male	1134 (44.4)	912 (43.5)	899 (42.6)	
Area					
	Rural	177 (6.9)	192 (9.2)	202 (9.6)	0.002
	Urban	2376 (93.1)	1905 (90.8)	1910 (90.4)	
Hospitalization Type					
	Emergency	857 (33.6)	667 (31.8)	781 (37.0)	<0.001
	Planned	1697 (66.4)	1430 (68.2)	1331 (63.0)	
Outcome					
	Died	79 (3.1)	74 (3.5)	57 (2.7)	0.003
	Discharged	2455 (96.1)	2020 (96.3)	2040 (96.6)	

* *p*-values were calculated using the Chi-square test or Fisher’s exact test, as appropriate.

**Table 2 healthcare-13-02520-t002:** Gender distribution based on hospitalization type.

Variable		Pandemic Period			*p* *
		Pre-Pandemic (n, %)	Wave 1 (n, %)	Wave 2 (n, %)	
Hospitalization Type					
Emergency	Female	489 (57.1%)	406 (60.9%)	511 (65.4%)	0.002
	Male	368 (42.9%)	261 (39.1%)	270 (34.6%)	
Planned	Female	931 (54.9%)	779 (54.5%)	702 (52.7%)	0.481
	Male	766 (45.1%)	651 (45.5%)	629 (47.3%)	

* *p*-values were calculated using the Chi-square test or Fisher’s exact test, as appropriate.

**Table 3 healthcare-13-02520-t003:** Comparison of referral sources by hospital during pre-pandemic and pandemic periods (H7 and KazIOR, N = 6763).

Referral Source	Pre-Pandemic	Wave 1		Wave 2		*p* *
	N (%)	N (%)	(%Change) ^†^	N (%)	(%Change) ^†^	
Ambulance	374 (14.6)	395 (18.8)	5.61	396 (18.8)	5.88	0.672
Another hospital	15 (0.6)	274 (13.1)	1726.67	250 (11.8)	1566.67	<0.001
Consultative and diagnostic	159 (6.2)	797 (38.0)	401.26	854 (40.4)	437.11	<0.001
Primary healthcare	508 (19.9)	376 (17.9)	−25.98	351 (16.6)	−30.91	<0.001
Self-referral	170 (6.7)	97 (4.6)	−42.94	159 (7.5)	−6.47	<0.001
Referrals from hematologists’ offices	1328 (52.0)	158 (7.5)	−88.1	102 (4.8)	−92.32	<0.001

^†^ The percentage change was calculated using the ‘Pre-pandemic’ period as the reference; * Chi-square test.

**Table 4 healthcare-13-02520-t004:** Final diagnosis organized by the number of visits and bed days in H7 and KazIOR (N = 6763).

Final Diagnosis	Number of Visits (Mean)			Bed Days (Mean)		
	Pre-Pandemic	Wave 1	Wave 2	Pre-Pandemic	Wave 1	Wave 2
Acute Leukemia	6	6	5	20	16	14
Anemia	6	6	3	13	12	10
Hemolytic Anemia	6	6	4	15	14	10
Lymphoproliferative Disorders	6	6	5	15	12	11
Malignant Disorders	16	5	4	9	9	8
Myeloproliferative Disorders	5	5	3	19	15	14
Other Neoplasms	5	5	4	16	9	13
Other Peripheral Arterial Diseases	3	15	-	17	11	-
Pathology of Hemostasis	6	6	3	16	13	12
*p* *	<0.001			0.034		

* The ANOVA test.

**Table 5 healthcare-13-02520-t005:** Final diagnosis by outcomes in H7 and KazIOR (N = 6763).

Final Diagnosis	Died, N (%)			Discharged, N (%)		
	Pre-Pandemic	Wave 1	Wave 2	Pre-Pandemic	Wave 1	Wave 2
Acute Leukemia	45 (37.8)	48 (40.3)	26 (21.8)	656 (39.2)	497 (29.7)	522 (31.2)
Anemia	3 (37.5)	1 (12.5)	4 (50.0)	101 (36.7)	95 (34.5)	79 (28.7)
Hemolytic Anemia	2 (66.7)	0 (0.0)	1 (33.3)	37 (44.0)	20 (23.8)	27 (32.1)
Lymphoproliferative Disorders	16 (36.4)	15 (34.1)	13 (29.5)	1301 (36.6)	1144 (32.2)	1109 (31.2)
Malignant Disorders	1 (50.0)	1 (50.0)	0 (0.0)	2 (25.0)	4 (50.0)	2 (25.0)
Myeloproliferative Disorders	11 (44.0)	4 (16.0)	10 (40.0)	195 (38.0)	141 (27.5)	177 (34.5)
Other Neoplasms	0 (0.0)	0 (0.0)	0 (0.0)	15 (53.6)	7 (25.0)	6 (21.4)
Other Peripheral Arterial Diseases	0 (0.0)	0 (0.0)	0 (0.0)	14 (100.0)	0 (0.0)	0 (0.0)
Pathology of Hemostasis	1 (33.3)	2 (66.7)	0 (0.0)	110 (34.1)	108 (33.4)	105 (32.5)
*p* *	0.300			<0.001		

* *p*-values were calculated using the Chi-square test or Fisher’s exact test, as appropriate.

## Data Availability

The data presented in this study are available on request from the corresponding author due to the privacy of the patients reason.

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
