# Peer review of "Impact of the COVID-19 Pandemic on Hemato-Oncology Services: A Retrospective Dual-Center Cohort Study in Kazakhstan"

_healthcare, 2025, doi:10.3390/healthcare13192520_

Round 1

Reviewer 1 Report

Comments and Suggestions for Authors

This manuscript deals with an interesting subject. It has, however, shown multiple gaps that have affected its quality:

  1. The main concern is related to the results presentation, which is very simplistic, while the interpretation is more complex. The presentation of figures 1 and 2 is not adapted. It is more adequate to present the results as bars or tables than as an evolution since these results did not really present a trend or an evolution. For table 4, it is better to delete the last column (transferred) since most of the results are equal to 0. In table 3 you should specify the number of visits (per day? Week?...). You should also add the numbers in table 2 (wave 1 &2). Regarding the interpretation, the authors should delete the introductory sentences and provide the results directly and succinctly to allow the reader to understand. The comparison of the number of deaths is also misleading. You should compare only the number of dead and discarded (the other numbers are very small, and nearly all of them are equal to zero).
  2. The second concern is related to the discussion, which is very superficial and is composed mostly of self-sentences and paragraphs, and the explanation is in some parts far from the content of the manuscript. The number of references is very limited (7 references only), which confirms my observation, and the comparisons are very superficial.
  3. The third concern is related to the introduction, which is very simplistic and very short with practically non-correlated sentences and paragraphs, and the sequencing of the ideas should be revised. For example, in the first paragraph, the authors should begin with COVID-19 in the world, and then they can describe it in the country. They should provide and describe some studies conducted in the same objective (even rare) to allow them to describe the hypothesis.

Other "minor" remarks:

You do not need to write chi-square in the abstract: you can use the abbreviation

Study setting: describe more the two hospitals (number of beds, ….).

Table 2: delete the first line (H7 and KazIOR)

Revise table 4 as requested and correct the form of line 1.

Extend the limitations of the study

Revise the conclusion to include the most important results

Author Response

We would like to thank the reviewer for the thorough evaluation and valuable feedback. In response, we have made several revisions to the manuscript to address the comments provided. Below, we outline our detailed responses, and the corresponding changes made.

Reviewer 1

-The main concern is related to the results presentation, which is very simplistic, while the interpretation is more complex. The presentation of figures 1 and 2 is not adapted. It is more adequate to present the results as bars or tables than as an evolution since these results did not really present a trend or an evolution. For table 4, it is better to delete the last column (transferred) since most of the results are equal to 0.

Response: Figures 1 and 2: We carefully considered the suggestion to replace Figures 1 and 2 with bar charts or tables. After testing both formats, we found that while the bar charts provided a structured presentation, they became visually congested and harder to interpret compared to the current format. We therefore retained the original format of Figure 1, which more clearly illustrates the distribution of hospitalization days. However, for Figure 2, we revised the visualization to improve clarity and ensure that the trends in repeated visits are more easily interpretable, while aligning better with the reviewer’s suggestion.

Table 4: We agree with the reviewer’s point. Since the “Transferred” category contained mostly zeros and added little interpretive value, we have removed this column from the revised version of Table 4 to streamline the presentation and avoid potential misinterpretation.

-In table 3, you should specify the number of visits (per day? Week?...).

Response: In Table 3, the “Number of visits” values represent the mean number of visits per patient during each study period (not per day or week). “Bed-days” indicate the mean number of hospital days per patient during the respective period. We have clarified this in a note below Table 3.

- You should also add the numbers in table 2 (wave 1 &2).

Response: We thank Reviewer 1 for this helpful suggestion. We respectfully chose not to add raw numbers to Table 2 because percentage change inherently represents relative differences between values across time periods, rather than absolute quantities. The absolute counts for referral sources during Wave 1 and Wave 2 are already provided in Table 1. Adding them again in Table 2 would introduce redundancy and potentially confuse readers. By focusing Table 2 solely on proportional shifts, we aim to provide a clearer representation of relative changes in referral dynamics across the pandemic periods.

-Regarding the interpretation, the authors should delete the introductory sentences and provide the results directly and succinctly to allow the reader to understand.

Response: We appreciate this suggestion. Introductory sentences have been minimized in the Results section to ensure that findings are presented more directly and succinctly.

-The comparison of the number of deaths is also misleading. You should compare only the number of dead and discarded (the other numbers are very small, and nearly all of them are equal to zero).

Response: We thank the reviewer for highlighting this point. In our revision, we focus mortality comparisons only on deaths and discharges, as transfer and unauthorized discharge counts were too small for meaningful interpretation. This clarification is now reflected in the Results section.

-The second concern is related to the discussion, which is very superficial and is composed mostly of self-sentences and paragraphs, and the explanation is in some parts far from the content of the manuscript.

Response: We have thoroughly revised the discussion to ensure it is more focused, evidence-based, and directly linked to the results presented in our study.

-The number of references is very limited (7 references only), which confirms my observation, and the comparisons are very superficial.

Response: In the revised manuscript, we have substantially expanded the number of references to strengthen the contextualization of our findings and provide a more comprehensive comparison with existing literature.

-The third concern is related to the introduction, which is very simplistic and very short with practically non-correlated sentences and paragraphs, and the sequencing of the ideas should be revised. For example, in the first paragraph, the authors should begin with COVID-19 in the world, and then they can describe it in the country. They should provide and describe some studies conducted in the same objective (even rare) to allow them to describe the hypothesis.

Response: The introduction has been substantially revised to provide a clearer structure. It now begins with the global impact of COVID-19 on healthcare and hemato-oncology services, includes evidence from international studies (e.g., EPICOVIDEHA, systematic reviews), and then transitions to the situation in Kazakhstan. The burden of hematologic malignancies globally and nationally has also been added to contextualize the study. Finally, we explicitly state the research aim and hypothesis at the end of the introduction to guide the reader.

-Other "minor" remarks:

-You do not need to write chi-square in the abstract: you can use the abbreviation.

Response: Abbreviated χ² in abstract.

- Study setting: describe more the two hospitals (number of beds, ….).

Response: added description of both hospitals (bed count, services).

-Table 2: delete the first line (H7 and KazIOR)

Response: removed from the first line in Table 2. 

- Revise Table 4 as requested and correct the form of line 1.

Response: corrected Table 4 line 1.

-Extend the limitations of the study.

Response: expanded limitations to address data biases.

-Revise the conclusion to include the most important results.

Response: revised the conclusion to summarize key results.

Reviewer 2 Report

Comments and Suggestions for Authors

The paper needs some editing. In particular, there are tense issues for certain words. For example, in line 77 the word increases is used and it should be increased. The editing necessary is not major just small things like this that need to be corrected. Additionally, there is some repetitive language, like in lines 106-108

In section 2.3, for variables included, why weren't income and education levels included? Please explain.

Table 1: Please breakdown the Hospitalization Type data by gender. Are there any statistical differences? This is important and needs to be included. 

In the discussion section, what are the policy implications of the research findings? Please add

Comments on the Quality of English Language

Minor editing necessary for tenses and repetitive language

Author Response

We would like to thank the reviewer for the thorough evaluation and valuable feedback. In response, we have made several revisions to the manuscript to address the comments provided. Below, we outline our detailed responses, and the corresponding changes made.

Reviewer 2

The paper needs some editing. In particular, there are tense issues for certain words. For example, in line 77 the word increases is used and it should be increased. The editing necessary is not major just small things like this that need to be corrected. Additionally, there is some repetitive language, like in lines 106-108

- In section 2.3, for variables included, why weren’t income and education levels included? Please explain.

Response: Thank you for this observation. We have reviewed the manuscript and corrected it.

- In section 2.3, for variables included, why weren’t income and education levels included? Please explain.

Response: Unfortunately, information on income and education levels was not available in our dataset. Therefore, we were unable to include these variables in our analysis.

Table 1: Please breakdown the Hospitalization Type data by gender. Are there any statistical differences? This is important and needs to be included.

Response:

We have now stratified the Hospitalization Type data by gender and conducted statistical comparisons. A new Table 2 has been added to the manuscript, presenting hospitalization type by gender, including the results of statistical testing.

-In the discussion section, what are the policy implications of the research findings? Please add

Response: We have added the policy implications of the research findings within the discussion section.

-Comments on the Quality of English Language

Minor editing necessary for tenses and repetitive language

Reviewer 3 Report

Comments and Suggestions for Authors

I read with interest the study presented by Yerdenova and colleagues on the Impact of the COVID-19 Pandemic on Hemato-Oncology Services: A Retrospective Dual-Center Cohort Study in Kazakhstan. The authors highlight two key findings: (1) the COVID-19 pandemic disrupted access to hemato-oncology services, leading to a measurable increase in emergency hospitalizations and mortality rates, and (2) there was a notable shift in referral pathways, with significant increases in referrals from other hospitals and emergency services, and sharp declines from hematologists’ offices. This topic fits well within the Healthcare journal’s aim and scope, offering insights relevant to its readership, particularly those concerned with health service delivery during crises.

Despite its merits, it is riddled with issues that need to be addressed before it can be accepted for publication. Below are nine major areas requiring revision, each accompanied by a request for clarification or change:

1- The introduction outlines the pandemic’s general effects but lacks a concise, overarching research question or hypothesis. The manuscript oscillates between describing service disruptions and reporting hospitalization patterns without clearly framing what it seeks to test or demonstrate. The authors should explicitly state a central research question and/or hypothesis at the end of the introduction to better guide the reader.

2- While the retrospective nature of the study is mentioned, there is insufficient discussion of the inherent biases (e.g., missing data, potential coding errors from electronic health records, and lack of control for confounders). Expand the methods or limitations sections to more transparently address these biases and their implications for interpreting the results.

3- The results section is saturated with percentages and p-values, but often lacks interpretation or connection to the broader context. For example, changes in referral sources are stated but not meaningfully explained. Integrate interpretive commentary within the results or discussion, clarifying the clinical significance of observed changes (e.g., why consultative referrals spiked or why ambulance referrals remained flat).

4- The authors mention using chi-square, Fisher’s exact, t-tests, and Mann-Whitney U, but there is no rationale for test selection, no description of assumptions checked, and no adjustment for multiple comparisons despite the numerous tests performed. Provide a clearer step-wise justification for the chosen tests, indicate how assumptions were verified, and discuss whether corrections for multiple testing (e.g., Bonferroni) were considered, as explained in this study (10.3390/medicina60101694).

5- The discussion selectively references studies from Europe and Korea but lacks a balanced comparison with wider literature on hemato-oncology care disruptions during COVID-19, particularly from similar healthcare systems or low- and middle-income countries. Enrich the discussion by incorporating relevant global studies (for instance, those from Eastern Europe or Central Asia) to better position Kazakhstan’s experience within a broader context.

6- Terms such as “Wave 1” and “Wave 2” are used without explicitly defining how these waves were determined epidemiologically, and the grouping of diagnoses (e.g., “malignant disorders” vs. “myeloproliferative disorders”) is not always consistent or clear. Clearly define all terms and classifications in the methods, perhaps with a supplementary table clarifying ICD-10 groupings and wave timelines.

7- The manuscript reports mortality changes (e.g., 3.5% in Wave 1 vs. 2.7% in Wave 2) but stops short of exploring why mortality decreased in Wave 2 despite ongoing service disruptions. Expand the discussion to consider plausible explanations (e.g., better clinical adaptations, improved COVID-19 management, COVID-19 vaccination (10.7759/cureus.33042) or changes in case mix), 

8- Although the study states that the ethics committee approved it and data were anonymized, there is no mention of data quality assurance, how missing data were handled, or whether data linkage between the two hospitals introduced inconsistencies. Provide more detailed reporting on data handling, including how missing or inconsistent data were addressed.

Comments on the Quality of English Language

The manuscript is largely understandable but contains awkward phrasing, redundancies, and formatting inconsistencies (e.g., “Wave 1” sometimes appears with a dash, sometimes without). These issues detract from readability. Undertake thorough language editing for clarity, remove redundancies, and standardize terminology and formatting throughout.

Author Response

We would like to thank the reviewer for the thorough evaluation and valuable feedback. In response, we have made several revisions to the manuscript to address the comments provided. Below, we outline our detailed responses, and the corresponding changes made.

Reviewer 3

I read with interest the study presented by Yerdenova and colleagues on the Impact of the COVID-19 Pandemic on Hemato-Oncology Services: A Retrospective Dual-Center Cohort Study in Kazakhstan. The authors highlight two key findings: (1) the COVID-19 pandemic disrupted access to hemato-oncology services, leading to a measurable increase in emergency hospitalizations and mortality rates, and (2) there was a notable shift in referral pathways, with significant increases in referrals from other hospitals and emergency services, and sharp declines from hematologists’ offices. This topic fits well within the Healthcare journal’s aim and scope, offering insights relevant to its readership, particularly those concerned with health service delivery during crises.

Despite its merits, it is riddled with issues that need to be addressed before it can be accepted for publication. Below are nine major areas requiring revision, each accompanied by a request for clarification or change:

1- The introduction outlines the pandemic’s general effects but lacks a concise, overarching research question or hypothesis. The manuscript oscillates between describing service disruptions and reporting hospitalization patterns without clearly framing what it seeks to test or demonstrate. The authors should explicitly state a central research question and/or hypothesis at the end of the introduction to better guide the reader.

Response: Thank you for this observation. We modified the research statement.

2- While the retrospective nature of the study is mentioned, there is insufficient discussion of the inherent biases (e.g., missing data, potential coding errors from electronic health records, and lack of control for confounders). Expand the methods or limitations sections to more transparently address these biases and their implications for interpreting the results.

Response: In the methods section we include sentence: There was no missing data for tested parameters and to ensure no coding errors during the data extraction two researchers (M.Y. and A.I.) performed de-identification process and specific code placement.

3- The results section is saturated with percentages and p-values but often lacks interpretation or connection to the broader context. For example, changes in referral sources are stated but not meaningfully explained. Integrate interpretive commentary within the results or discussion, clarifying the clinical significance of observed changes (e.g., why consultative referrals spiked or why ambulance referrals remained flat).

Response: We added additional literature to the discussion section and discussed probable changes and disruptions in patients flow regarding hospitalization and treatment.

4- The authors mention using chi-square, Fisher’s exact, t-tests, and Mann-Whitney U, but there is no rationale for test selection, no description of assumptions checked, and no adjustment for multiple comparisons despite the numerous tests performed. Provide a clearer step-wise justification for the chosen tests, indicate how assumptions were verified, and discuss whether corrections for multiple testing (e.g., Bonferroni) were considered, as explained in this study (10.3390/medicina60101694).

Response: In methods section we explained how we tested normality distribution for continuous variables.

5- The discussion selectively references studies from Europe and Korea but lacks a balanced comparison with wider literature on hemato-oncology care disruptions during COVID-19, particularly from similar healthcare systems or low- and middle-income countries. Enrich the discussion by incorporating relevant global studies (for instance, those from Eastern Europe or Central Asia) to better position Kazakhstan’s experience within a broader context.

Response: Thank you for this comment. We have added several references and accordingly discussed in discussion section.

6- Terms such as “Wave 1” and “Wave 2” are used without explicitly defining how these waves were determined epidemiologically, and the grouping of diagnoses (e.g., “malignant disorders” vs. “myeloproliferative disorders”) is not always consistent or clear. Clearly define all terms and classifications in the methods, perhaps with a supplementary table clarifying ICD-10 groupings and wave timelines.

7- The manuscript reports mortality changes (e.g., 3.5% in Wave 1 vs. 2.7% in Wave 2) but stops short of exploring why mortality decreased in Wave 2 despite ongoing service disruptions. Expand the discussion to consider plausible explanations (e.g., better clinical adaptations, improved COVID-19 management, COVID-19 vaccination (10.7759/cureus.33042) or changes in case mix)

Response: We included in the discussion section proposed concerns.

8- Although the study states that the ethics committee approved it and data were anonymized, there is no mention of data quality assurance, how missing data were handled, or whether data linkage between the two hospitals introduced inconsistencies. Provide more detailed reporting on data handling, including how missing or inconsistent data were addressed.

Response: We had no missing data and two researchers are allocated to collect analyze and import data in order to secure proper data flow. There was no inconsistency of the data between two hospitals.

Comments on the Quality of English Language

The manuscript is largely understandable but contains awkward phrasing, redundancies, and formatting inconsistencies (e.g., “Wave 1” sometimes appears with a dash, sometimes without). These issues detract from readability. Undertake thorough language editing for clarity, remove redundancies, and standardize terminology and formatting throughout.

All revision need to be incorporated in manuscript with red letters.

Round 2

Reviewer 1 Report

Comments and Suggestions for Authors

The authors made considerable efforts to improve the quality of the manuscript. However, while some comments were not taken into consideration, other concerns still exist.

  1. Even though the first and the second paragraphs are well structured, the ideas of paragraphs 3 and 5 are fragmented and have no correlation with the content of the first paragraphs and even with the objectives of the study. For example, you passed directly from hematology to covid-19 in the country without any correlation. Try to revise, please. In addition, you didn't cite any research done on this topic in the region.

A concern that bothered me is the fact that table 3 contains the same results of table 1 but with different statistical analysis (results). It is better to include these results in table 3 since they were not described anywhere in table 1. Also, they are not standardized (other in table 1, from the hematologist office in table 3). Revise, please.

While figure 2 was corrected, the form of figure 1 is a real concern for me, and I consider that it is not adapted (I understand your concern). You should, however, find the best consensus to better interpret your results—(the figure is also of poor quality)

Delete sentence line 258-260

Delete the sentence of line 279 "the last column….equal to 0). Readers will not know that you have deleted this column

Indicate the tables or figures one time only in the text (ex: table 5 is indicated 3 times)

Another concern is the use of the term across the pandemic. I consider that this term is not adapted since the pre-pandemic period cannot be included in the pandemic period. Revise throughout the manuscript (including tables and figures), please.

These comments have not been taken into consideration:

Study setting: describe more the two hospitals (number of bed….)

Response: added description of both hospitals (bed count, services)

Not provided

-Extend the limitations of the study.

Response: expanded limitations to address data biases.

More details are required

-Revise the conclusion to include the most important results.

Response: revised the conclusion to summarize key results.

No corrections observed

At last, the manuscript requires professional editing for English and grammatical errors

Author Response

We sincerely thank reviewer 1 for their valuable observation and helpful feedback. We have made several changes to the manuscript to address reviewer’s comments. See responses and specific changes to each comment detailed below.

Reviewer 1 Round 2

  1. Even though the first and the second paragraphs are well structured, the ideas of paragraphs 3 and 5 are fragmented and have no correlation with the content of the first paragraphs and even with the objectives of the study. For example, you passed directly from hematology to covid-19 in the country without any correlation. Try to revise, please. In addition, you didn't cite any research done on this topic in the region.

Response:

Thank you for this important comment. We revised the Introduction to improve coherence between the paragraphs and ensure alignment with the study objectives by adding transition sentences linking global and national contexts. We also cited regional research (Ishkinin Y, et al.; Front. Oncol., 2025) and globally (Mohammadali Jafari et al; Hematology, Transfusion and Cell Therapy, 2022,) to strengthen the relevance of our work.

  1. A concern that bothered me is the fact that table 3 contains the same results of table 1 but with different statistical analysis (results). It is better to include these results in table 3 since they were not described anywhere in table 1. Also, they are not standardized (other in table 1, from the hematologist office in table 3). Revise, please.

Response:

Thank you for your observation. We have deleted the referral sources from Table 1 and included these results in Table 3, where they are now presented with the appropriate statistical analysis.

  1. While figure 2 was corrected, the form of figure 1 is a real concern for me, and I consider that it is not adapted (I understand your concern). You should, however, find the best consensus to better interpret your results—(the figure is also of poor quality)

Response:

Thank you for your valuable feedback. We have corrected Figure 1, improving clarity, and revised it to ensure better interpretation and presentation of the results.

  1. Delete sentence line 258-260

Delete the sentence of line 279 "the last column….equal to 0). Readers will not know that you have deleted this column

Indicate the tables or figures one time only in the text (ex: table 5 is indicated 3 times)

Response:

Thank you for your observations. We have deleted the sentences at lines 258–260 and the sentence at line 279 regarding the last column. In addition, we have revised the manuscript to ensure that each table and figure is referenced only once in the text.

  1. Another concern is the use of the term across the pandemic. I consider that this term is not adapted since the pre-pandemic period cannot be included in the pandemic period. Revise throughout the manuscript (including tables and figures), please.

Response:

Thank you for this important comment. We have carefully revised the manuscript, including the tables and figures, and replaced the term “across the pandemic” with more accurate wording, distinguishing clearly between the pre-pandemic and pandemic periods.

  1. These comments have not been taken into consideration:

Study setting: describe more the two hospitals (number of bed….)

Response:

Thank you for your observation. We have now described both hospitals in more detail, including the number of beds, and incorporated this information into the Study Design and Data Collection section.

  1. Not provided

-Extend the limitations of the study.

More details are required

-Revise the conclusion to include the most important results.

Response:

Thank you for your valuable feedback. We have extended the Limitations section by providing more details on potential biases, generalizability, and data coverage. In addition, we have revised the Conclusion to highlight the most important findings of the study.

  1. No corrections observed

At last, the manuscript requires professional editing for English and grammatical errors

Response:

Thank you for your comment. We have carefully revised the manuscript for English language, grammar, and style. If further professional editing is required, we will consider using the journal’s recommended language editing service.

Reviewer 3 Report

Comments and Suggestions for Authors

Well done to the authors for their efforts in addressing most of the raised concerns. 

While the manuscript has been significantly improved, a few minor issues remain:

  • Given the wide journal's readership, the presentation of the statistical analysis remains unclear and confusing, especially for those without statistical knowledge/background. Authors are strongly advised to adopt the approach presented in this study (10.3390/medicina60101694), and should be referenced to avoid scientific plagiarism.
  • The figures are not supplemented with appropriate figure legends, only titles. Authors should provide concise figure legend for each figure.
  • The discussion section is incredibly long and difficult to follow. Authors should divide their discussion section into reasonable paragraphs, each containing a single idea/concept.

Author Response

We sincerely thank reviewer 1 for their valuable observation and helpful feedback. We have made several changes to the manuscript to address reviewer’s comments. See responses and specific changes to each comment detailed below.

Reviewer 3 Round 2

  1. Given the wide journal's readership, the presentation of statistical analysis remains unclear and confusing, especially for those without statistical knowledge/background. Authors are strongly advised to adopt the approach presented in this study (10.3390/medicina60101694) and should be referenced to avoid scientific plagiarism.

Response:

Thank you for your valuable comment. We have revised the statistical analysis in the manuscript and have taken into consideration the approach presented in the suggested study (10.3390/medicina60101694).

  1. The figures are not supplemented with appropriate figure legends, only titles. Authors should provide concise figure legend for each figure.

Response:

Thank you for your comment. We have now provided concise and informative figure legends for all figures to ensure clarity and proper interpretation of the results.

  1. The discussion section is incredibly long and difficult to follow. Authors should divide their discussion section into reasonable paragraphs, each containing a single idea/concept.

Response:

Thank you for your valuable feedback. We have revised the Discussion section by dividing it into clear, coherent paragraphs, each focusing on a single idea or concept, to improve readability and flow.